# Local Stratopause Temperature Variabilities and their Embedding in the Global Context

Ronald Eixmann[1], Vivien Matthias[2], Josef Höffner[1], Gerd Baumgarten[1], and Michael Gerding[1]

[1]Leibniz Institute of Atmospheric Research, Schloss-Strasse 6, 18225 Kühlungsborn, Germany
[2]Potsdam Institute for Climate Impact Research, Telegrafenberg A 31, 14473 Potsdam, Germany

**Correspondence:** Ronald Eixmann (eixmann@iap-kborn.de)

**Abstract.** The stratopause is by definition the transition between the stratosphere and mesosphere. During winter the circulation at mid and high latitudes in the stratosphere is mainly driven by quasi stationary planetary waves (PWs) while the circulation in the mesosphere is mainly driven by gravity waves (GWs). The question arises whether PWs or GWs dominate the variability of the stratopause. The most famous and dramatic variability of the middle atmosphere is a sudden stratospheric warming (SSW) generated by PWs interacting with the polar vortex. A similar phenomenon but smaller in magnitude and more regional are stratopause temperature enhancements (STE) initially observed by local measurements and generated by breaking PWs. Thus it seems that PWs dominate the variability of the stratopause. In this study we want to quantify to which amount quasi stationary PWs contribute to the stratopause variability. To do that we combine local lidar observations at Kühlungsborn ($54°N, 11°E$) and Andenes ($69°N, 16°E$) with global MERRA-2 reanalysis data bringing the local variability of the stratopause into the global context. Therefore we compare the temperature time series at Kühlungsborn and Andenes at $2\,hPa$, the altitude where STEs maximize, with characteristics (amplitude and phase) of PWs with wave numbers 1, 2 and 3. We found that for Kühlungsborn and Andenes 98% of the local day-to-day variability of the stratopause can be explained by the variability of PWs with wave number 1, 2, and 3. Thus, the winter stratopause day-to-day variability is highly dominated by the variability of PWs.

## 1 Introduction

Upper stratosphere/lower mesosphere (USLM) temperatures are a sensitive indicator for climate change (Rind et al., 1998). Especially during winter dynamical changes of the USLM influence the entire stratosphere (Hitchman et al., 1989) via the downward control principle (Haynes et al., 1991) and that in turn can affect our tropospheric weather and climate (Baldwin et al., 2001; Kodera et al., 2008; Kretschmer et al., 2018).

The stratopause represents by definition the transition between the stratosphere and mesosphere. During winter the polar stratosphere is characterised by a strong circumpolar westerly wind, known as the polar vortex. The polar vortex is disturbed in the northern hemisphere by planetary Rossby waves (PWs) generated in the troposphere and propagating upward under westerly wind conditions into the middle atmosphere (Charney and Drazin, 1961; Matsuno, 1970). The strongest PWs that occur in the winter middle atmosphere are quasi stationary PWs (Matsuno, 1970). When those waves break they deposit their

momentum in the middle atmosphere (McIntyre and Palmer, 1983) and thus driving especially the stratosphere away from its radiative equilibrium by generating a mean meridional circulation from equator to pole with downward motion at the pole which adiabatically warms the polar stratosphere (e.g., Hitchman et al., 1989; Plumb, 2002; Manney et al., 2008). In the mesosphere the main driver of the circulation are gravity waves (GWs) by implementing a mean meridional circulation from
the summer to the winter pole with accompanied upward motion in summer and downward motion in winter resulting in a warmer winter mesosphere than expected from the radiative equilibrium (Fritts and Alexander, 2003). Thus the variability of the stratopause region apart from the climatological mean state seems to depend on the variability of PWs and GWs.

The most famous disturbance of the winter polar middle atmosphere triggered by PWs is the sudden stratospheric warming (SSW) which is characterized by a dramatic warming of the stratosphere and a simultaneous cooling of the mesosphere (Mat-
suno, 1971; Andrews et al., 1987; Matthias et al., 2012). A similar phenomenon that is smaller in magnitude and more regional is the "stratopause temperature enhancement (STE)" (Meriwether and Gerrard, 2004) also known as "stratopause warming" (Braesicke and Langematz, 2000) or "USLM disturbance" (Manney et al., 2008; Thayer et al., 2010; Greer et al., 2013). While the term USLM disturbance often refers to the whole 3-D structure of the disturbance, the term STE refers to the observable regional temperature increase of the stratopause. In this study we will use the term STE since we focus on the variability of the
stratopause temperature here.

Greer et al. (2013) investigated the mean characteristics of STE by constructing a climatology of these events using 20 years of stratospheric assimilated data from the U.K. Meteorological Office (UKMO). They found that STE occur between November and March in the northern hemisphere during the winter season with a pronounced preference in December and preferentially over Northeastern Russia and Scandinavia. The mean duration of a STE is 8 days. In earlier observations STE occurred always
shortly before a SSW (von Zahn et al., 1998) but later on there was evidence that STE do not necessarily develop into a SSW (Meriwether and Gerrard, 2004; Manney et al., 2008). Greer et al. (2013) showed that in their assimilated model data all major SSW were preceded by a STE while that was the case only for half of the minor SSW. Approximately one third of all STE did not develop into a SSW at all. Similar to SSW, there is often a mesospheric cooling around $75\,km$ associated with the STE (Meriwether and Gerrard, 2004; Thayer et al., 2010). Until now it is not completely understood how STEs develop. Several
authors investigated the formation of STE using models (Fairlie et al., 1990; Braesicke and Langematz, 2000), reanalysis data (Greer et al., 2013) or global satellite data (Thayer et al., 2010). The common sense is that PWs interact with the polar vortex in the upper stratosphere leading to an ageostrophic circulation driving vertical motions resulting in adiabatic heating in the upper stratosphere and cooling in the mesosphere due to the induced change in the vertical propagation of gravity waves. There is evidence that this perturbation may grow through baroclinic instability (Fairlie et al., 1990; Thayer et al., 2010; Greer et al.,
2013) which is fed off by strong upward propagating PWs between $10$ and $2\,hPa$ (Thayer et al., 2010; Greer et al., 2013).

So in general the main cause for stratopause day-to-day variabilities seem to be PWs. In this study we want to quantify to which amount quasi stationary PWs contribute to the stratopause day-to-day variability at one location. Therefore we combine local lidar measurements with global reanalysis data. Lidar observations are performed at Andenes ($69°N, 16°E$) and Küh-lungsborn ($54°N, 11°E$). Andenes is located mostly within the polar vortex and Kühlungsborn at the edge of the polar vortex
(see Greer et al. (2013), their Fig. 5). These two locations are especially suited for this kind of study since they lie in the area

where STE can occur (Greer et al., 2013). MERRA-2 reanalysis data will be used to investigate the temporal variability of the stratopause of the last 39 winters (1980/81 - 2018/19). The paper is structured as follows: in section 2 the two lidar systems and the MERRA-2 reanalysis data are briefly described as well as the methods applied in this study. Our results are described in section 3, discussed in section 4 and summarized in section 5.

## 2 Data and methods

To bring the local variability of the stratopause into the global context we combine local lidar measurements at two locations at high and mid-latitudes with global reanalysis data of MERRA-2. The local lidar measurements are used to characterize STEs and to roughly estimate the quality of local MERRA-2 profiles. Using the temporal evolution of MERRA-2 reanalysis data at these two locations and the corresponding latitudes, the variability at one location is compared with the global variation of stationary PWs regarding their amplitude and phase.

### 2.1 Local lidar observations

In this study we mainly use Rayleigh/Mie/Raman (RMR) lidar systems which are located at the ALOMAR observatory near Andenes in the Norwegian Arctic ($69°N, 16°E$) and at Kühlungsborn in Germany ($54°N, 11°E$). The climatology of each station comprises ten years of measurements between 2002 and 2012 in winter between December and February. For the calculation of the temperature from the RMR lidar raw data we use the hydrostatic density integration as described, e.g., by Chanin and Hauchecorne (1981), based on the backscatter at $532\,nm$ wavelength.

The ALOMAR RMR lidar is able to obtain temperatures from above the stratospheric aerosol layer (i.e. above $\sim 22 - -34\,km$ depending on actual conditions) up to $\sim 90\,km$ during both darkness and daylight conditions. The lidar is actually a twin lidar system with two independent lasers ($\sim 20\,W$ average power at $532\,nm$) and two tiltable telescopes of $1.8\,m$ diameter. Typical integration times for this study are $1\,h$ at $2.5\,km$ vertical resolution. Further details about the ALOMAR RMR lidar are described by Baumgarten (2010) and Schöch et al. (2008).

The RMR lidar at Kühlungsborn is similar to the system at ALOMAR. An additional detection channel for $N_2$ vibrational Raman backscatter at $608\,nm$ allows for the correction of stratospheric aerosol effects. By this, the lowest altitude bins for Rayleigh temperature retrieval can be set to $22\,km$. The $\sim 20\,W$ average power at $532\,nm$ and the four receiver telescopes of $60\,cm$ diameter each allow for $1\,km$ vertical resolution after $1\,h$ integration. Typically, the upper altitude limit is above $85\,km$ altitude. In this study the data are limited to nighttime conditions. The RMR lidar data at Kühlungsborn is complemented by temperature profiles obtained by a collocated potassium resonance lidar. This technique provides absolute temperatures between $\sim 85$ and $105\,km$ by probing the Doppler broadening of the D1 resonance line of potassium atoms that exist in this altitude range. These data are also used as a start value for the hydrostatic temperature retrieval for the Kühlungsborn RMR lidar, while the calculation for the ALOMAR RMR lidar is initialised with climatological data. Further details of the temperature lidars at Kühlungsborn are described by Alpers et al. (2004) and Gerding et al. (2008).

Note that in this study only nighttime measurements (Andenes: 134, Kühlungsborn: 71) with at least three hours of measurement are taken into account from both lidar systems. A nighttime mean of the lidar temperature profiles is calculated to minimize the impact of small scale effects.

## 2.2 Global MERRA-2 reanalysis data

To set local stratopause variabilities into the global context we use global reanalysis data from the Modern-Era Retrospective analysis for Research and Applications Version 2 (MERRA-2; Gelaro et al. (2017); Bosilovich et al. (2015); Molod et al. (2015)). Here we use daily means of the 3-hourly instantaneous output on 42 constant pressure levels ranging from $1000\,hPa$ to $0.1\,hPa$, i.e. from the surface up to $68\,km$. The horizontal resolution is $0.625$ degree in longitude and $0.5$ degree in latitude. For our analysis we use MERRA-2 reanalysis data from winter 1980/81 to 2018/19, i.e. 39 winters in total whereas the term

winter includes the months December, January and February. Note that MERRA-2 assimilates temperature and ozone profiles from MLS satellite in the upper stratosphere and mesosphere starting in August 2004 (Gelaro et al., 2017). Therefore the stratopause temperatures are more reliable after that (Gelaro et al., 2017) which makes MERRA-2 particularly suited for our analysis.

## 2.3 Methods

Here we use the regional variability STE to investigate the agreement between the local lidar measurements and the global MERRA-2 reanalysis data. Similar to Thayer et al. (2010) and Greer et al. (2013) we identify a STE if the temperature profile is at least $15\,K$ above the climatological mean at $2\,hPa$ (near the stratopause). To calculate the deviation from the climatological mean for the lidar nighttime temperatures the NRLMSISE-00 empirical model of the atmosphere (Picone et al., 2002) is used as a reference atmosphere. This model comprises an all-embracing day-by-day temperature climatology. Thus the deviation

from the climatological mean for the lidar temperature data is calculated by subtracting the NRLMSISE-00 model climatology from the nighttime lidar measurement for each altitude, day and location separately. Consequently we can divide the lidar nighttime temperatures into STE and non-STE. Note that for lidar data the STE criterion is applied exactly at the stratopause altitude and thus varying with the height of the stratopause while for MERRA-2 data a fixed pressure level of $2\,hPa$ is used as in Thayer et al. (2010) and Greer et al. (2013). This is done due to the relatively low number of lidar measurements with

strongly varying stratopause heights and for the comparability of our MERRA-2 results with other studies.

The deviation from the climatological mean in MERRA-2 data is calculated by removing the long-term mean computed from the 39 winters of MERRA-2 data, for each day, altitude and longitude separately at the latitudes where Kühlungsborn and Andenes are located. Note that daily mean values are used for MERRA-2 data and nighttime means are used for lidar data. Thus an impact of tides and long period gravity waves especially on the lidar data can not be excluded.

To bring the local values at Andenes and Kühlungsborn into the global context regarding PWs, the three predominant zonal wave numbers 1, 2 and 3 are fitted to the MERRA-2 data at both latitudes separately. The following description is illustrated in Fig. 1: for each day the original daily mean data (blue line) at each latitude at $2\,hPa$ is decomposed into the three dominating wave numbers 1, 2 and 3 (green line for wave 1, orange line for wave 123, i.e. all three waves together) using a

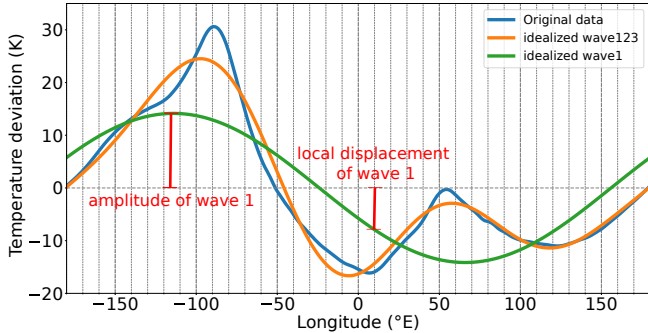

**Figure 1.** Illustration of the decomposition of the original time series into wave number 1, 2 and 3 and of the concept of amplitude and local displacement which denotes the value at a specific longitude.

least-squares procedure. Thus we somewhat reconstructed the original data based on the different PW components. Here we call the maximum absolute value of a wave the *amplitude* of that wave and the value between zero and an arbitrary location on the wave the *local displacement*. In this study the local displacement is at $11°E$ for Kühlungsborn and $16°E$ for Andenes respectively. With the help of these two concepts (amplitude and local displacement) not only the change in the amplitude of a
PW but also the change in phase of the wave is taken into account. For example, the amplitude of a PW could remain the same, but changes in phase still can change the value of the local displacement at a given location.

Following this approach we reconstruct the whole time series for Andenes $(69°N, 16°E)$ and Kühlungsborn $(54°N, 11°E)$ based on PWs with wave numbers 1, 2 and 3. To visualize the differences and similarities of the original time series and the reconstructed time series we correlate them for each location separately (i.e.Andenes and Kühlungsborn) as will be shown in
the next section.

## 3   Results

Before we compare the temporal evolution of local stratopause temperatures with the temporal evolution of PWs at the same altitude and latitude, we study the vertical profile of STEs using lidar data and compare the results with profiles of MERRA-2 reanalysis data.

**3.1   Characteristics of vertical profiles of STEs from lidar measurements**

Figure 2 shows all available winter nighttime mean profiles between 2002 and 2012 for Andenes (a) and Kühlungsborn (b) and those identified as STE (c, d). Profiles in c) and d) are a subset of a) and b), respectively. There is a measurement rate of 13% for Andenes and 7% for Kühlungsborn of all winter days for this period. If one compares the long-term mean profiles (blue line) of Andenes and Kühlungsborn in Fig. 2 a) and b), it is noticeable that the altitude of the stratopause is $4\,km$ lower at
Kühlungsborn $(47\,km)$ than at Andenes $(51\,km)$. The stratopause altitude during a STE is about $3\,km$ lower at Kühlungsborn

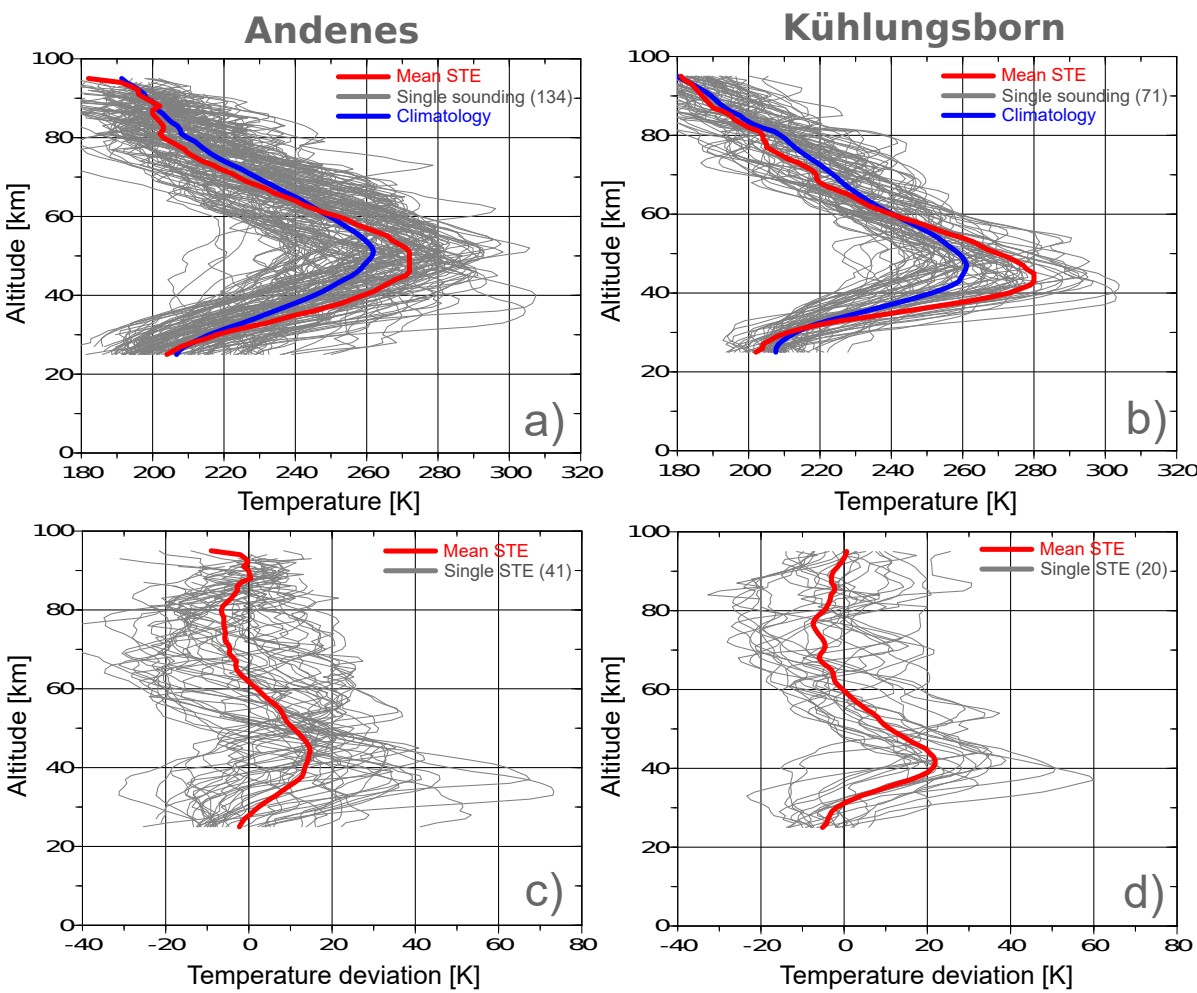

**Figure 2.** a+b) Vertical profiles (grey lines) of all nighttime lidar temperature measurements between 2002 and 2012 at Andenes (a) and Kühlungsborn (b) and their respective long-term mean (blue line). c+d) All detected STE profiles (grey lines) with subtracted long-term winter mean derived from lidar measurements. The red line represents in all plots the respective mean STE profile.

and $7\,km$ lower at Andenes than the corresponding mean stratopause altitude. Possible causes for the difference in height of the stratopause between mid- and high latitudes as well as between STE and the long-term mean are discussed in section 4. For Andenes 41 out of the 134 measured profiles are classified as STE (30%) while for Kühlungsborn 20 out of 71 measured profiles are classified as STE (28%). Note that we will call the period 2002 – 2012 the "lidar period" in the following.

## 3.2 Vertical profiles of STEs from MERRA-2 reanalysis data

Figure 3 a) and b) show vertical profiles of STEs (grey lines) and their average (red line) from MERRA-2 reanalysis data between winter 1980/81 and 2018/19 as well as the temperature climatology for all available 39 winters (blue line). The climatological stratopause altitude (cf. blue line) is $57\,km$ at Andenes and about 49 km at Kühlungsborn and thus higher compared to the lidar data at both locations ($50\,km/47\,km$ respectively). However similar to the lidar data the stratopause is in general higher at Andenes than at Kühlungsborn. As already mentioned in section 2, MLS observations are assimilated in MERRA-2 starting in summer 2004 improving the upper stratosphere and lower mesosphere (Gelaro et al., 2017). We will call the period 20005 – 2019 the "MLS period" in the following. Restricting the analysed period to the MLS period results in a climatological stratopause altitude of $53\,km$ at Andenes and $48\,km$ at Kühlungsborn (see Supplement). Thus the difference between lidar observations and MERRA-2 reanalysis decreases for Andenes and even vanishes for Kühlungsborn.

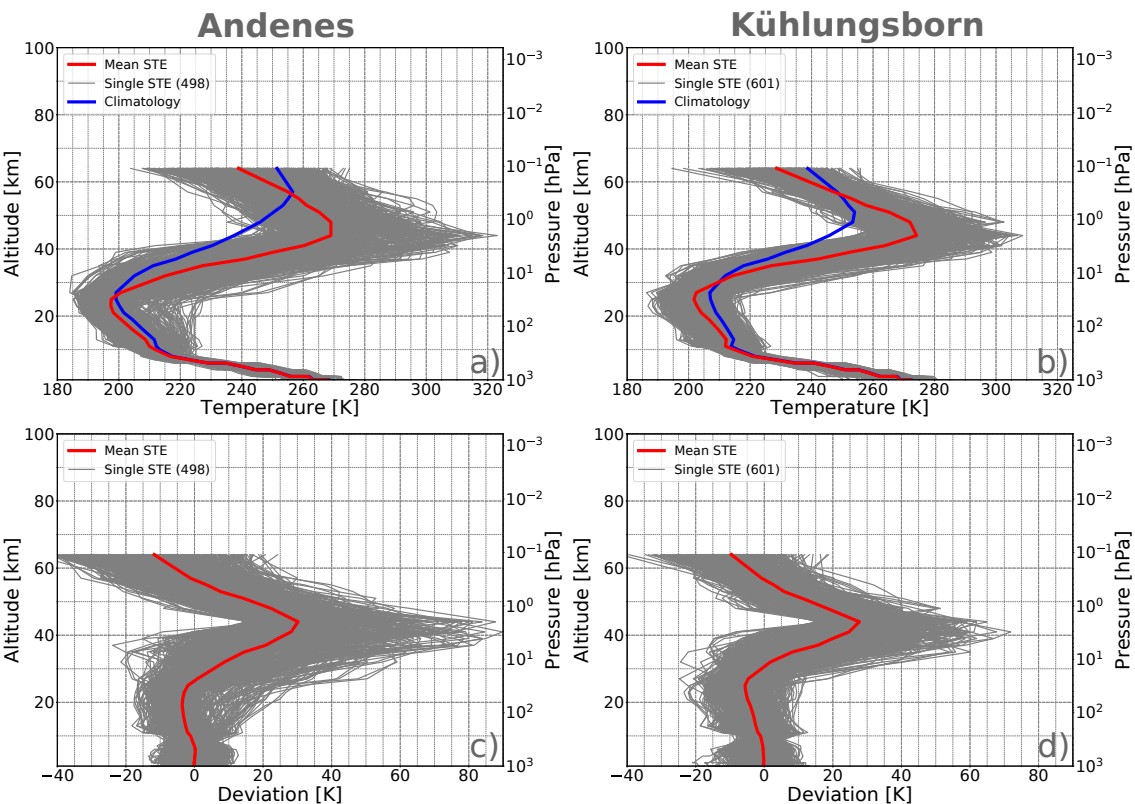

**Figure 3.** a+b) Vertical profiles (grey) of all STEs between winter 1980/81 and 2018/19 in MERRA-2 data at Andenes and Kühlungsborn and their respective mean (red line). The blue line represents the climatology over all winter profiles. c+d) Same as in the top row but climatology (blue line) is subtracted.

The reason why we define STEs at 2 hPa can be seen in Fig. 3 c) and d), showing again vertical profiles of STEs but with subtracted climatology. In general a STE ranges between $35$ and $55\,km$ with its maximum around $44\,km$ which is in good agreement to findings of Meriwether and Gerrard (2004) and Greer et al. (2013) and to the lidar data (see Fig. 2 c+d). The maximum mean STE temperature differs between lidar and MERRA-2 data by about $2\,K$ at Andenes and $4\,K$ at Kühlungsborn. The STE occurrence rate is $14\%$ at Andenes and $17\%$ at Kühlungsborn. Differences between MERRA-2 and lidar data are discussed in the following.

### 3.3 Comparison between lidar observations and MERRA-2 reanalysis data

Figure 4 compares the absolute mean STE profiles at Andenes and Kühlungsborn derived from lidar observations (blue) and MERRA-2 data (red). At Andenes the difference of the absolute peak temperature between lidar and MERRA-2 is $3\,K$ at $2\,hPa$ while it is $6\,K$ at Kühlungsborn. In general, the MERRA-2 profiles are colder than the lidar profiles indicating that small-scale disturbances like gravity waves are not well represented in MERRA-2.

There are quite large differences in the STE occurrence rates between lidar ($30\%/\,28\%$) and MERRA-2 ($14\%/17\%$). However, when we apply the STE criterion ($15\,K$ above climatological mean) not to the fixed pressure level of $2\,hPa$ but similar to the lidar data to the varying stratopause height then the STE occurrence rates derived from MERRA-2 are $27\%/\,23\%$ and thus much closer to the lidar STE occurrence rates.

Note that the differences further decrease when we additionally restrict MERRA-2 to the lidar period ($30\%/23\%$) and that we get slightly lower rates when we restrict to the MLS period ($25\%/18\%$). Thus differences in the STE occurrence rates between lidar and MERRA-2 vanish at Andenes and decrease at Kühlungsborn when restricting to the lidar period. The remaining differences between lidar and MERRA-2 can be attributed for one thing to the different statistical basis which is much better for the MERRA-2 data than for the lidar data since there are 296/230 STE profiles in the MERRA-2 data restricted to the lidar period but only 41/20 in the lidar data at Andenes/Kühlungsborn. On the other hand small-scale dynamics like gravity waves are not perfectly represented in MERRA-2 in the upper stratosphere which could explain some part of the differences between lidar and MERRA-2. A further discussion on the influence of small-scale dynamics can be found in section 4. Differences of the STE occurrence rates between the lidar period and MLS period can also occur due a year-to-year variability of the STE occurrence.

Further causes for the above described differences between lidar measurements and MERRA-2 reanalysis data are the higher temporal and vertical resolution of the lidar measurements at higher altitudes in comparison to MERRA-2 data as well as the different length of the daily averaging interval. While lidar profiles are nighttime averages, daily means are calculated for the MERRA-2 reanalysis data. Thus lidar temperature profiles are biased by solar tides and gravity waves with longer periods. When averaging nighttime measurements with a length of at least 3 hours, then only a part of a tide or large period gravity wave is averaged out. This effect is much weaker in MERRA-2 data since a daily mean is calculated here. Thus especially the bias from solar tides and long period gravity waves is much weaker since the entire oscillation is used in the averaging process. Note that we checked the STE occurrence rates using only nighttime MERRA-2 data and found, depending on which

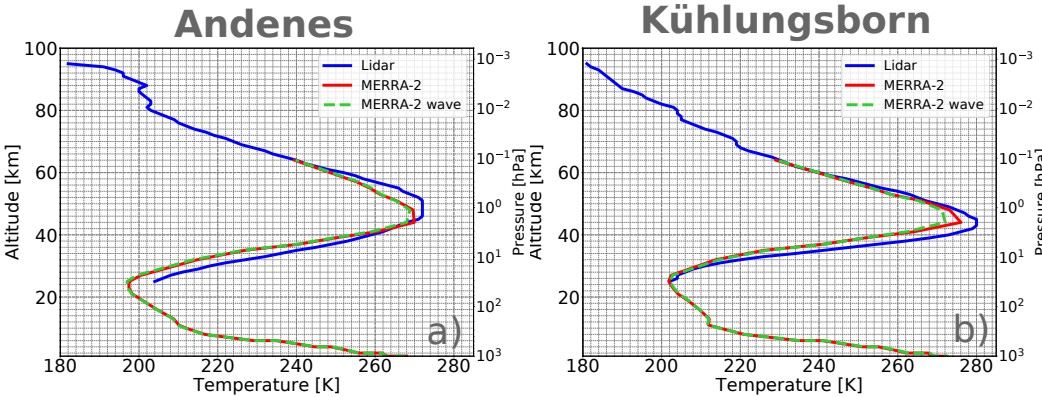

**Figure 4.** Mean vertical STE profiles at Andenes (a) and Kühlungsborn (b) derived from lidar observations (blue), MERRA-2 (red) and MERRA-2 reconstructed for wave 123 (green, dashed).

part of the night is used, increased occurrence rates by up to $3\%$ at both locations. Thus the lidar STE occurrence rates might be slightly lower when daily mean data would be available.

Even though there are differences in magnitude of STEs between lidar and MERRA-2, they are in good agreement regarding the structure of the STEs, hence MERRA-2 reanalysis data can be used to bring local observations into the global context.

## 3.4 Local variability embedded into the global context

To bring local variability into the global context we compare the time series at Andenes and Kühlungsborn) at $2\,hPa$ with the time series from the amplitude of wave numbers 1, 2, 3 and 123 (all together) at the same latitude and with the time series of the local displacement for this very location (also for wave number 1, 2, 3 and 123).

Figure 5 shows the temperature deviation from the climatological daily mean of the original time series at one location at $2\,hPa$ (blue line), the time series of the amplitude for the respective PWs (green line) and the reconstructed time series (orange line) based on the respective PW (local displacement) for winter 2009/10. This is done for Andenes (left column) and Kühlungsborn (right column) and for the different wave numbers (each row represents a different wave number) using MERRA-2 data. The embedding of the local variability into the global context is exemplary shown for winter 2009/10. A similar analysis for all available 39 winters can be found in the Supplement.

First, focusing on wave 123 for Andenes and Kühlungsborn, it is evident that the blue (original time series) and the orange line (reconstructed time series) lay almost perfectly on top of each other. The correlation coefficient of the original and reconstructed time series for winter 2009/10 is 0.99 for Andenes and 0.97 for Kühlungsborn. Thus $99\%$ of the day-to-day variability of the temperature at Andenes can be explained by the variability of PWs with wave number 1, 2 and 3 in MERRA-2 data. The correlation of the original time series with the amplitude (green line) results in correlation coefficients of almost zero for Andenes and only $0.28$ for Kühlungsborn. The discrepancy between the correlation coefficients of the amplitude and local dis-

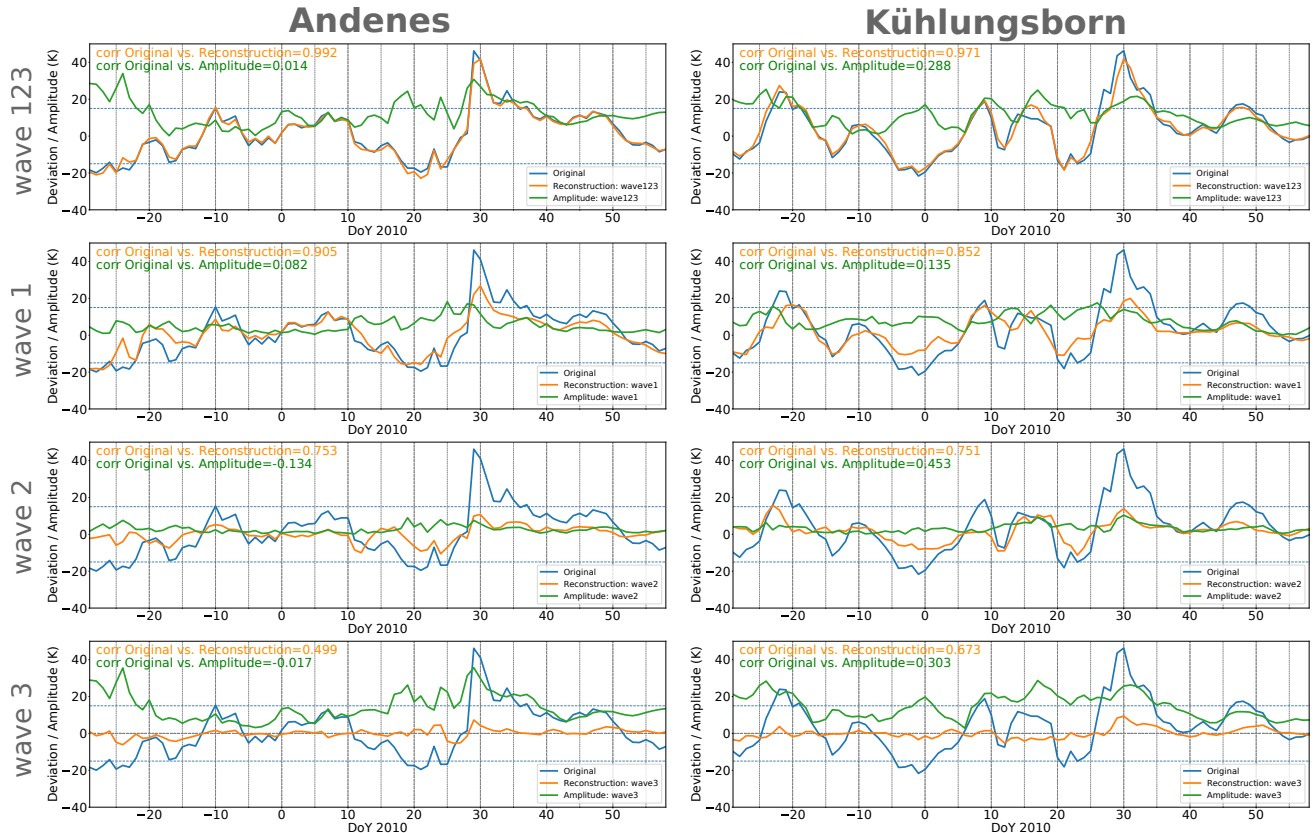

**Figure 5.** Temporal evolution of the temperature deviation from the climatological daily mean at Andenes (left) and Kühlungsborn (right) at $2\,hPa$ (blue line). The amplitude of each individual wave number (different rows) is represented by the green lines while the local displacement, also called reconstructed time series, is represented by the orange lines. The horizontal grey dashed lines mark the positive and negative threshold of a STE. All data presented here are derived from MERRA-2 data.

placement results from the fact that the amplitude describes only the increase and decrease of the amplitude of the respective PW while the local displacement also takes the change in phase into account. This can be observed especially well around day 0 in Kühlungsborn (upper row in Fig. 5). While the amplitude increases, the value of the local displacement decreases dramatically, due to the shift in phase of the waves. Around day -10, Kühlungsborn lies under the positive part of the wave packet where the temperature maximises. In the next 10 days, the phase shifts in such a way that Kühlungsborn is now located under the minimum half of the wave packet where the temperature minimises. The amplitude is blind to such a phase shift event explaining its much lower correlation coefficient. During other occasions, for example between day -10 and 10 at Andenes, the amplitude and local displacement values are much more synchronous pointing to a stationary phase. Due to the much better

|              | Andenes | Kühlungsborn |
|--------------|---------|--------------|
| wave 123     | 0.98    | 0.98         |
| wave 1       | 0.87    | 0.79         |
| wave 2       | 0.64    | 0.75         |
| wave 3       | 0.31    | 0.5          |

**Table 1.** Correlation coefficients between the original time series and the reconstructed time series for all available winters for Andenes and Kühlungsborn.

description of the local variability by the local displacement or reconstructed time series (orange line) we will focus on this in the following.

Considering the individual wave numbers the largest correlation coefficient between the original time series and the reconstructed time series occurs for wave 1 at both locations (Andenes: $r_A = 0.9$, Kühlungsborn: $r_K = 0.84$). Thus wave number

1 dominates the variability of the winter stratopause. In both stations wave 2 ($r_A = 0.75, r_K = 0.74$) is slightly less important than wave 1 ($r_A = 0.9, r_K = 0.84$). Wave 3 plays by far the smallest role ($r_A = 0.5, r_K = 0.65$).

Similar to the example winter 2009/10 described in detail above, we correlated the original time series of all available winters with the reconstructed time series of every individual wave number as well as all wave numbers together. The individual correlation coefficients for both locations can be found in table 1. In general 98% of the daily stratopause variability at Andenes

and Kühlungsborn can be explained by the variability of PWs with wave number 1, 2 and 3. Also the other characteristics of the individual wave numbers found in winter 2009/10 can be generalized. Note that we conducted this analysis also to temperature time series at $1\,hPa$, the climatological stratopause altitude at Kühlungsborn, and received the same results (not shown).

## 4   Discussion

As shown in Fig. 2 and 3 the stratopause is about $4\,km$ lower at Kühlungsborn than at Andenes in the climatological mean.

This phenomenon of an elevated and warmer winter stratopause at high latitudes compared to mid-latitudes was also observed by Hitchman et al. (1989) using global satellite measurements. Combining these global satellite observations with a 2-D model they found evidence that the elevated polar winter stratopause is caused by GWs driving a meridional circulation with downwelling over the winter pole. The mean residual meridional circulation is characterised by downwelling in the middle atmosphere resulting in adiabatic warming and differs between the stratosphere and mesosphere in its driving mechanism.

While it is driven by PWs in the stratosphere (e.g., Andrews et al., 1987; Plumb, 2002), it is driven by GWs in the mesosphere (e.g., Lindzen, 1981; Plumb, 2002). The stratopause is by definition the transition between these two atmospheric layers. However, Haynes et al. (1991) showed that the mean meridional circulation at any level is determined by the vertically integrated momentum forcing above that level. Garcia and Boville (1994) confirmed the significant contribution of GWs in the mesosphere on the circulation in the stratosphere at high latitudes by using a simple numerical model in the middle atmosphere with

parametrized GW and PW breaking. Removing the GW breaking in their model results in a significant colder stratosphere and

less downwelling at high latitudes during winter. Thus the difference in the stratopause altitude between mid- and high latitudes can be explained by the mean meridional circulation in the high latitudes mesosphere driven by GWs .

The altitude of the stratopause is lower during STEs than in the climatology for both locations (cf. Fig. 2 and 3). This decreased altitude of the stratopause during STEs was also observed by earlier studies (von Zahn et al., 1998; Meriwether and Gerrard, 2004). The formation of a STE and thus also the descent of the stratopause is explained by several authors (Fairlie et al., 1990; Thayer et al., 2010; Greer et al., 2013) as follows: interactions between the polar vortex and PWs can be associated by localized momentum forcing resulting in a synoptic ageostrophic circulation which is accompanied with a strong vertical motion. This vertical motion points downward in the upper stratosphere, adiabatically warming this region and therefore descending the stratopause. However, our study shows that a large part of the temporal evolution of temperature anomalies around the stratopause can be simply described by a superposition of conservatively evolving planetary waves (cf. Fig. 5), including STE events. There is also an upward pointing part of the vertical motion in the lower mesosphere adiabatically cooling this region at the same time. The theoretically described cooling in the lower mesosphere was, to the authors knowledge, first reported by Meriwether and Gerrard (2004) during a STE above Sondrestrom in Greenland ($67^\circ N, 309^\circ E$) in December 2000. This cooling can be confirmed for mean STE by our lidar measurements at both locations as well as by the MERRA-2 reanalysis data (cf. Fig. 2 and 3). This is in agreement with the study of Greer et al. (2013) investigating the mean characteristics of USLM disturbances. Note that the cooling is stronger at polar than at mid-latitudes, presumably because mid latitudes are if at all at the edge of the polar vortex and thus less affected because the downwelling is located in the center of the polar vortex.

Figure 4 shows the mean STE profiles of Andenes and Kühlungsborn derived from lidar observations (blue), MERRA-2 data (red) and MERRA-2 data reconstructed for wave 123 (green, dashed). In the following we assume that the difference between lidar and the reconstructed MERRA-2 data set is solely caused by small-scale disturbances like gravity waves even though part of the differences can also have other reasons as discussed in section 3.3. Focussing on the STE definition altitude ($2\,hPa$) there is a difference of about $3\,K$ at Andenes and $8\,K$ at Kühlungsborn. Especially at Andenes this difference is small compared to the overall STE amplitude ($30\,K$, see Fig. 3c) indicating that PWs strongly dominate the STE development there. At Kühlungsborn the difference is about one third of the total STE amplitude ($28\,K$, see Fig. 3d) indicating that gravity waves play a much larger role in the STE development at mid latitudes than at high latitude in winter. Thus the impact of small-scales disturbances like gravity waves on the day-to-day variability of the stratopause is much larger at Kühlungsborn than at Andenes. Nevertheless, PWs dominate the STE development in winter at Kühlungsborn as well.

As shown and discussed before PWs dominate the day-to-day variability of the stratopause. However there are also small-scale processes like gravity waves that also contribute to the stratopause variability. Even though MERRA-2 does not capture all small-scale disturbances as discussed above, we will discuss in the following the general difference between the original time series at $2\,hPa$ and the reconstructed time series for wave 123 as for example shown in Fig. 5 assuming that the difference is caused by small-scale effects that actually are captured by MERRA-2. The standard deviation over all available winter between

the original time series and the reconstruced time series is $1.9\,K$ at Andenes and $2.76\,K$ at Kühlungsborn indicating again a stronger impact of gravity waves at mid- than at high latitudes. The maximum difference between these two time series is $18\,K$ at Andenes and even $25\,K$ at Kühlungsborn indicating that the STE development can also be dominated by smaller-scale disturbances as other studies showed before (Garcia and Boville, 1994; Meriwether and Gerrard, 2004; Thayer and Livingston, 2008).

The dominating role of PWs in the stratosphere has long been known (e.g., Andrews et al., 1987; Plumb, 1989; Rosenlof and Holton, 1993). Our study quantitatively shows their impact on local measurements. With this new knowledge and the concept of local displacement, local measurements can be better brought into the global context. Additionally, effects which are not based on PW variability can be much better identified and thus investigated. For example, in winter 1994/95 around day 25 (see

Supplement) there is a strong temperature enhancement at the stratopause where the original time series is about $17\,K$ higher than the reconstructed time series based on wave 1, 2 and 3. This STE lies shortly before a SSW but in contrast to other STEs, as for example those reported by von Zahn et al. (1998) and Meriwether and Gerrard (2004) (cf. Supplement), the agreement with the PW variability is much lower and thus other effects seem to play a role. However it is not the scope of this study to investigate or discuss this unknown effect but with the help of the local displacement concept it can be better identified.

**5   Summary**

In this study we wanted to quantify the amount of quasi stationary PWs contributing to the day-to-day variability of the stratopause in northern hemisphere winter. Therefore we combine local lidar measurements at mid- and high latitudes with global MERRA-2 reanalysis data to bring the local variability into the global context. With the help of stratopause temperature enhancements (STEs) it is shown that local lidar measurements at Kühlungsborn ($54°N, 11°E$) and Andenes ($69°N, 16°E$)

are in good agreement with global MERRA-2 reanalysis data around the stratopause. Note that this agreement is even better when focusing on the period where MLS data are assimilated in MERRA-2 especially regarding the climatological altitude of the stratopause (see Supplement).

Both, observations and reanalysis, show a lower stratopause at Kühlungsborn than at Andenes in their climatology as well as during STEs and a lower stratopause during STEs compared to its respective climatology. Mean STE profiles are similar in

structure in lidar observations and MERRA-2 data but differ in magnitude by about $3\,K$ at Andenes and $8\,K$ at Kühlungsborn. The STE occurrence rate is higher in the lidar observations as in the MERRA-2 data probably due to biases in the lidar nighttime mean and not well represented small-scale disturbances like gravity waves in MERRA-2.

Using MERRA-2 reanalysis data at the two designated latitudes and locations and applying the concept of local PW displacement it is shown that $98\%$ of the local day-to-day stratopause temperature variability at Andenes and Kühlungsborn can

be explained by the variability of global PWs with wave numbers 1, 2 and 3. Thus PWs highly dominate the day-to-day variability of the stratopause at mid- and high latitudes in the northern hemisphere in MERRRA-2. However the differences in the mean STE magnitude between lidar and MERRA-2 data indicate that PWs strongly dominate the STE development at

Andenes but are less dominant at Kühlungsborn where around one third of all STE might be generated by gravity waves and other small-scale disturbances.

*Data availability.* MERRA-2 data are freely available from the MERRA project at https://gmao.gsfc.nasa.gov/reanalysis/MERRA-2/. Lidar data are available from the corresponding author upon request.

5   *Author contributions.* RE and VM led the study and wrote the paper. RE also contributed to the lidar data collection and analysis, and to the interpretation of the results. VM analysed the MERRA-2 data, performed the wave analysis and interpreted the results. GB is responsible for and analysed the lidar measurements at Andenes. MG is responsible for lidar measurements at Kühlungsborn. JH provided the database for the lidar measurements. All authors read and approved the final manuscript.

*Competing interests.* The authors declare that they have no conflict of interest.

10   *Acknowledgements.* We thank Axel Gabriel for very helpful discussions. We would like to thank the people at the National Aeronautics and Space Administration (NASA, USA) for providing MERRA-2 data. This work was partly supported by the DFG project 218499286 (MS-GWaves/PACOG) which is part of the DFG Research Unit FOR 1898 (MS-GWaves).

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
