# Peer review of "Local Stratopause Temperature Variabilities and their Embedding in the Global Context"

_Annales Geophysicae, 2019_

## Referee Comment (RC1) · Anonymous Referee #1 · 26 Sep 2019

This paper presents a study on the contribution of planetary waves (PWs) to the local stratopause temperature variability based. It is based on the use of global MERRA-2 analysis to estimate the contribution of PWs wavenumber 1, 2 and 3 at fixed locations where Rayleigh lidar observations are available, Andenes at polar latitude in Northern Norway and Kühlungsborn at middle altitude in Northern Germany. In the first part of the results section the authors compare the stratopause characteristics at these two llocations retrieved from lidar observations and from MERRA-2 analysis. Two cases are considered, the overall wintertime stratopause climatology and the climatology of stratopause temperature enhancements (STEs). The rest of the paper is based only on the use of MERRA-2 data to estimate the contribution of PW wavenumbers 1, 2 and 3 to the local stratopause variability. Although I consider that this subject may be

interesting, I don't think that this paper brings new interesting information because it does not address the subject the right way as explained below.

The main reason is that the estimation of the contribution of PW 1, 2 and 3 to the total stratopause temperature variability is made using only MERRA-2 data on not taking advantage of having more resolved local lidar profiles. The reanalysis smooth out the small-scale perturbations that can increase the variability, including local temperature perturbations induced by gravity waves breaking and PWs with high wave number. Furthermore there are very few observations assimilated in the model at the stratopause altitude and in the mesosphere. This is not surprising that most of the variability in the reanalysis comes from the PWs with lowest wavenumbers but this does not prove that it is the same in the reality. It shows only that MERRA-2 analysis captures mostly the contribution of larger scale PWs. It would have been much more interesting to use the MERRA-2 analysis to compute the PWs contribution to the stratopause temperature at lidar locations and to remove this contribution to stratopause temperature observed by the lidars. However this would imply that MERRA-2 reproduces faithfully the large scale temperature variability.

The comparison of the STE characteristics from lidar observations and MERRA-2 analysis made in sections 3.1 and 3.2 is also not convincing. I don't consider that the differences are small as it is claimed at line 8, page 6. For instance, at Andenes, there is a 7-km difference between the climatological stratopause altitude in MERRA-2 analysis (57 km) and in lidar observations (50 km). This is not at all a small difference. A careful comparison of average temperature profiles and stratopause characteristics should have been done. This is also a prerequisite to use MERRA-2 data for embedding the local observations in the global context.

---

## Referee Comment (RC2) · Anonymous Referee #2 · 11 Oct 2019

The study by Eixmann et al. is an interesting work that investigates stratopause temperature enhancements (STEs) that occur at mid and high latitudes in the Northern Hemisphere during winter. The paper focuses on two locations, Kühlungsborn and Andenes, where lidar data are available. It is found that the characteristics of STEs seen in lidar data are similar to those found in MERRA-2 reanalysis data, and it is concluded that the mechanism that causes STEs can be investigated using MERRA-2 data.

For further analysis, planetary waves 1-3 are determined from MERRA-2 data, and local temperature anomalies caused by those waves and their superposition are calculated for the two selected locations. This is a novel approach that helps to interpret measurements at fixed locations. It is found that almost all STEs in MERRA-2 can be explained by the variability of planetary waves with wavenumber 1, 2, and 3, which is

an interesting and new result.

The paper is well written and of interest for the readership of Annales Geophysicae. Publication of the paper is therefore recommended after minor revisions.

Main comments:

(1) It is quite encouraging to find that many STEs can be explained already by superposition of planetary waves. However, there is a remaining difference in STE occurrence rates between lidars and MERRA-2 that is not explained. Therefore possible effects of smaller scale dynamics, likely gravity waves, should be discussed in more detail.

Reviewer 1 suggested to calculate the local contribution of MERRA-2 planetary waves at the times and locations of the lidar profiles to estimate the contribution of breaking gravity waves and higher wavenumber planetary waves from the observations. In this way, it could be determined for the lidar observational record how many STEs would be expected by superposition of planetary waves 1-3. The difference between this number and the total lidar STE occurrence rate could then be used as an estimate for the effect of smaller scale dynamics. It could then also be checked whether this difference is in agreement with the difference between lidar STE occurrence rates and the MERRA-2 occurrence rates based on more years.

(2) As also mentioned by Reviewer 1, it would be desirable to have better constrained reanalysis data in the stratopause region. Starting with the year 2004 MERRA-2 assimilates MLS data. For this later period MERRA-2 should be one of the best freely available reanalysis data sets in the stratopause region. Therefore it would be interesting to additionally perform the MERRA-2 analysis only for the period starting with winter 2004/2005 to find out whether there is an effect on the MERRA-2 results, and whether the agreement between lidar and MERRA-2 climatologies changes and perhaps even improves.

Specific comments:

(1) p.2, l.2: The reference Hitchman et al. (1989) should be mentioned also here because this is the earlier work.

(2) p.2, l.5/6: Did you mean the variability of the stratopause, here? Or the mean state? Or both?

(3) p.4, l.5-10: Please add the information that, different from most other reanalyses, MERRA-2 assimilates MLS satellite soundings in the stratosphere and the mesosphere, starting with 2004. Therefore stratopause temperatures in MERRA-2 should be more reliable after that.

(4) p.4, l.23: As averages as short as 3 hours are used for the lidar data also long period gravity waves may not completely average out. This may add some noise to the lidar STE statistics.

(5) Fig.2 seems to be in disorder: Firstly, the panels are not labeled (a) ... (d). Secondly, the caption is not correct: All nighttime measurements are displayed in gray, not blue, and their mean in blue, not red. Thirdly, it looks like the same average STE curve (in red) is used for both Andenes and Kühlungsborn. This does not make much sense because there are temperature offsets between both stations. Fourthly, the legend in (a) and (b) it should read "single soundings", not "single STE". Fifthly, please use English titles for the different panels.

(6) p.5, l.13-16: Please note that in Fig.2 the same red curve is by mistake used in all panels. Therefore please check numbers after correction!

(7) p.6 after l.22: You should add some discussion on the differences between MERRA-2 and lidar STE occurrence rates.

The STE occurrence rate of MERRA-2 at the two locations is 14% and 17%, while for lidar these rates are 41/134=31% and 20/71=28%, about twice as high.

This difference is quite remarkable and shows that large scale dynamics like planetary waves that are well represented in MERRA-2 explain about 50% of the STEs, which is

an encouraging result, but are likely not the only mechanism causing STEs. Obviously, also smaller scale dynamics like gravity waves contribute to the lidar STE statistics.

(8) p.6, l.27: Here you write: "Figure 4 shows the original time series at one location at 2 hPa (blue line), ..." In the caption of Fig.4 it is mentioned that the blue line would be the "temperature deviation from the climatological daily mean". This information should also be given in the text of the manuscript. It should also be mentioned in the text that MERRA-2 data are shown.

(9) p.8, l.21/22: As has been shown in Fig.4, most of the temporal evolution of temperature anomalies can be described by a superposition of planetary waves, including the STE events.

This sparks the question whether "localized momentum forcing resulting in a synoptic ageostrophic circulation" is needed to explain the STEs. It seems that STEs can be caused already by superposition of conservatively evolving planetary waves.

(10) p.8/p.9: In the Discussion and the Summary it should be mentioned that the STE occurrence rates derived from lidar are twice as high as for MERRA-2 which indicates that dynamics at smaller scales can also lead to STEs.

Other comments:

(1) p.1, l.1: During winter the circulation -> During winter the circulation at mid and high latitudes

(2) p.2, l.17: in the northern hemisphere winter -> in the northern hemisphere during the winter season

(3) p.2, l.33: The word "whereby" does not fit here. Suggest to start new sentence:

(54N, 11E) whereby Andenes... -> (54N, 11E). Andenes...

(4) p.3, l.1: winter -> winters

(5) p.3, l.1: structure -> structured

(6) p.4, l.9: winter -> winters

(7) p.4, l.18: measuerement -> measurement

(8) p.4, l.32: in phase of the wave. -> in phase of the wave is taken into account.

(9) p.4, l.32/33: suggested rewording for the sake of comprehensibility:

For example, the amplitude of a PW remains the same but changes in phase and thus can change the value of the local displacement at Andenes. -> For example, the amplitude of a PW could remain the same, but changes in phase still can change the value of the local displacement at a given location.

(10) p.5, l.12: (red line) -> (blue line)

(11) p.5, l.19: the climatology -> the temperature climatology

(12) p.5, l.19: winter -> winters

(13) caption of Fig.3: the long-term mean is subtracted. -> the long-term mean (blue line) is subtracted.

(14) p.8, l.5: polar stratopause -> polar winter stratopause

(15) caption of Fig.3: Külungsborn -> Kühlungsborn

---

## Author Comment (AC1) · 28 Nov 2019

**General information:** We found a small bug in our MERRA-2 data reading routine. For every month that was read by the routine the last day of the month was missing. Thus there were three days missing each year. We fixed this. Note that our analysis is **not** affected by this bug. There are only a few more STE events now because there are three more days per year analyzed (39*3=117).

This paper presents a study on the contribution of planetary waves (PWs) to the local stratopause temperature variability based. It is based on the use of global MERRA-2 analysis to estimate the contribution of PWs wave number 1, 2 and 3 at fixed locations where Rayleigh lidar observations are available, Andenes at polar latitude in Northern Norway and Kühlungsborn at middle altitude in Northern Germany. In the first part of the results section the authors compare the stratopause characteristics at these two llocations retrieved from lidar observations and from MERRA-2 analysis. Two cases are considered, the overall wintertime stratopause climatology and the climatology of stratopause temperature enhancements (STEs). The rest of the paper is based only on the use of MERRA-2 data to estimate the contribution of PW wave numbers 1, 2 and 3 to the local stratopause variability. Although I consider that this subject may be interesting, I don't think that this paper brings new interesting information because it does not address the subject the right way as explained below.

The main reason is that the estimation of the contribution of PW 1, 2 and 3 to the total stratopause temperature variability is made using only MERRA-2 data on not taking advantage of having more resolved local lidar profiles. The reanalysis smooth out the small-scale perturbations that can increase the variability, including local temperature perturbations induced by gravity waves breaking and PWs with high wave number. Furthermore there are very few observations assimilated in the model at the stratopause altitude and in the mesosphere. This is not surprising that most of the variability in the reanalysis comes from the PWs with lowest wave numbers but this does not prove that it is the same in the reality. It shows only that MERRA-2 analysis captures mostly the contribution of larger scale PWs. It would have been much more interesting to use the MERRA-2 analysis to compute the PWs contribution to the stratopause temperature at lidar locations and to remove this contribution to stratopause temperature observed by the lidars. However this would imply that MERRA-2 reproduces faithfully the large scale temperature variability.

We only agree partly with the reviewer. To get PW characteristics out of local lidar profiles one would need at least five lidar stations ideally evenly distributed around one latitude. Those five lidar stations have to measure simultaneously just to identify unambiguously wave number 1. The amount of lidars needed to characterize higher wave numbers is even much higher. This is of cause desirable but does not reflect reality. Just because of this restriction not to do the study would definitely not help to better understand the interaction between local and global disturbances. On the contrary, with the help of this study local disturbances can be much better classified as real local disturbances caused by small scale dynamics or as phase shifts of planetary waves. Distinguishing between these two processes is essential. Thus to put local measurements into the global context using global reanalysis data is the best way to do this at the moment.

PW with higher wave numbers are very unlikely in the (upper) stratosphere due to the Charney-Drazin criterion which implies that PWs can only propagate upward in westerly winds that are not too strong. The upper critical limit depends on the zonal wave number, thus the critical zonal wind strength decreases with increasing wave number. For example, it is mandatory that the zonal wind is below ~10m/s for PWs with wave number 4 to propagate into the stratosphere. Thus synoptic-scale waves with wave numbers higher than wave number 3 can only propagate into the stratosphere under very special condition, for example during SSWs. However under normal winter condition the zonal wind at middle and polar latitudes is too strong (20 – 60m/s) and thus synoptic-scale waves cannot propagate to the stratopause.

The reviewer is right, it is possible that local temperature perturbations like STEs can also be induced by small-scale dynamics like gravity waves which might not be captured by MERRA-2. However, as we showed in our study, PWs dominate the STE development. Note that an additional analysis and discussion on the impact of small-scale dynamics on the day-to-day variability of the stratopause region (P12L19 – P13L3) reveal that the impact of small-scale dynamics seems to be larger at Kühlungsborn than at Andenes. We added a much more detailed analysis and discussion on differences between lidar observations and MERRA-2 data in the new subsection 3.3 (P8L7- P9L2) including a new figure (now Fig. 4) directly comparing mean STE profiles derived from lidar and MERRA-2 for both locations.

Starting in 2004, MERRA-2 assimilates MLS data in the upper stratosphere and lower mesosphere improving MERRA-2 output in that region (Gelaro et al., 2017; their Figure 21). To test the impact of MLS assimilation in MERRA-2 on our results we rerun our analysis restricting to the MLS period but got very similar results. A detailed discussion on that can be found in our answers to Reviewer 2 (her/his main comment 1 and 2). We added a discussion on this topic in section 2.2 and 3.3 including a comparison of climatology and mean STE profiles in the Supplement.

Furthermore there are very few observations assimilated in the model at the stratopause altitude and in the mesosphere. The comparison of the STE characteristics from lidar observations and MERRA-2 analysis made in sections 3.1 and 3.2 is also not convincing. I don't consider that the differences are small as it is claimed at line 8, page 6. For instance, at Andenes, there is a 7-km difference between the climatological stratopause altitude in MERRA-2 analysis (57 km) and in lidar observations (50 km). This is not at all a small difference. A careful comparison of average temperature profiles and stratopause characteristics should have been done. This is also a prerequisite to use MERRA-2 data for embedding the local observations in the global context.

The reviewer is right. The difference in stratopause altitude between the lidar and MERRA-2 is not small especially in Andenes. We corrected the respective text passages and as Reviewer 2 suggested, we performed our MERRA-2 analysis also for the period starting in 2004/05 when MLS is assimilated to MERRA-2 which should improve the quality around the stratopause. Firstly, we compare the vertical profiles of the climatology and mean STE from the complete data set with the period starting in winter 2004/05 (see Figure 1 below) for both locations.
In Andenes the stratopause altitude of the climatological vertical profile is about 5km lower in the MLS period (blue dashed line) than in the complete MERRA-2 data set (blue solid line). The difference to the lidar measurements (green dashed line) is now 2km which is much better compared to the completely available data set. In Kühlungsborn the stratopause altitude is slightly decreased by 2km in the MLS period compared to the completely available period. However the difference to the lidar measurements is now 1km which can be attributed to the vertical resolution of MERRA-2 in that altitude region. Thus the climatological profiles of MERRA-2 in the MLS period and lidar are very similar for Kühlungsborn.
However the difference between both climatological profiles of all available data and of the MLS period is very small at 2hPa the pressure level (see Figure 1 below) where STEs are defined here. We added a much more detailed discussion on the differences between lidar observation and MERRA-2 data set in section 3.3 and in the Discussion.

[Figure]

*Figure 1 Vertical profiles of the climatologies (blue) based on the whole MERRA-2 data set (solid) and on the period starting with winter 2004/05 (dashed) as well as the mean STE profiles (red) again based on the whole MERRA-2 data set (solid) and on the period starting with winter 2004/05 (dashed). The green dashed line represents the lidar climatology.*

---

## Author Comment (AC2) · 28 Nov 2019

**General information:** We found a small bug in our MERRA-2 reading routine. For every month that was read by the routine the last day of the month was missing. Thus there were three days missing each year. We fixed this. Note that our analysis is not affected by this bug. There are only a few more STE events now because there are three more days per year (39*3=117).

The study by Eixmann et al. is an interesting work that investigates stratopause temperature enhancements (STEs) that occur at mid and high latitudes in the Northern Hemisphere during winter. The paper focuses on two locations, Kühlungsborn and Andenes, where lidar data are available. It is found that the characteristics of STEs seen in lidar data are similar to those found in MERRA-2 reanalysis data, and it is concluded that the mechanism that causes STEs can be investigated using MERRA-2 data. For further analysis, planetary waves 1-3 are determined from MERRA-2 data, and local temperature anomalies caused by those waves and their superposition are calculated for the two selected locations. This is a novel approach that helps to interpret measurements at fixed locations. It is found that almost all STEs in MERRA-2 can be explained by the variability of planetary waves with wave number 1, 2, and 3, which is an interesting and new result.
The paper is well written and of interest for the readership of Annales Geophysicae. Publication of the paper is therefore recommended after minor revisions.

**Main comments:**
(1) It is quite encouraging to find that many STEs can be explained already by superposition of planetary waves. However, there is a remaining difference in STE occurrence rates between lidars and MERRA-2 that is not explained. Therefore possible effects of smaller scale dynamics, likely gravity waves, should be discussed in more detail. Reviewer 1 suggested to calculate the local contribution of MERRA-2 planetary waves at the times and locations of the lidar profiles to estimate the contribution of breaking gravity waves and higher wave number planetary waves from the observations. In this way, it could be determined for the lidar observational record how many STEs would be expected by superposition of planetary waves 1-3. The difference between this number and the total lidar STE occurrence rate could then be used as an estimate for the effect of smaller scale dynamics. It could then also be checked whether this difference is in agreement with the difference between lidar STE occurrence rates and the MERRA-2 occurrence rates based on more years.

The reviewer is right, the discussion of differences between lidar observations and MERRA-2 data especially regarding small-scale disturbances was only shortly discussed in the paper. We changed the structure of section 3 (Results) a little bit and added a subsection including a new figure (new Fig. 4) showing a direct comparison between the mean STE profile derived from lidar observations and MERRA-2 data (P8L7 - P9L2). Additionally we added two paragraphs about the differences and their possible causes in the Discussion (P12L19 – P13L3).
The numbers of the respective STE occurrence rates are now added in the text (P6L3/4 and P8L4/5) as well as a more detailed discussion on possible causes for the differences (P8L11 – P9L2).
As suggested by you and Reviewer 1 we added a figure showing the difference between the mean STE profile derived from lidar observations in comparison to the mean STE profile derived from the reconstructed MERRA-2 time series (wave 123, green dashed line in Fig. 4). We added a more detailed discussion on the impact of small-scale disturbances like gravity waves on the STE development and the differences between Andenes and Kühlungsborn (P12L19-28). Note that we still emphasize that planetary waves dominate the STE development even though there might be occasions when gravity waves dominate the STE development (discussed in P12L30-P13L3).
Note that synoptic-scale waves with higher wave numbers (wave number >=4) are very unlikely in the stratosphere since they cannot propagate if the zonal wind is too strong (Charney-Drazin criterion). For example, in order that wave 4 can propagate upward, the zonal wind is not allowed to exceed 10m/s which is always the case in the winter stratosphere expect during a SSW. Higher wave

numbers have even a lower zonal wind threshold making it even more unlikely to propagate into the stratosphere. Thus synoptic-scale waves cannot propagate from the troposphere into the stratosphere under normal winter conditions.

We think that the differences in the STE occurrence rates between lidar and MERRA-2 have multiple reasons: firstly, lidar observations are biased since we only have nighttime means. A test with MERRA-2 using different length of nighttime data resulted in up to 3% higher occurrence rates for MERRA-2 compared to daily mean analysis. Secondly, lidar data may be biased by their timing of observations. Since measurements are only possible under clear-sky conditions depending mostly on large-scale weather pattern which can, under certain conditions, propagate up into the stratosphere as PWs causing STEs. This way the STE occurrence rate of lidar measurements might appear larger than it actually is in reality. Thirdly, MERRA-2 may have weaknesses in representing small-scale disturbances and in gravity wave parametrization possibly leading to the lower STE occurrence rates. However, we tried to estimate the effect of small-scale dynamics on the stratopause variability under the assumption that the difference in the mean STE profiles derived by lidar and the reconstructed MERRA-2 time series is solely caused by these small-scale dynamics (P12L30-P13L3). Nevertheless we do not want to give exact numbers since we cannot rule out biases neither in the lidar nor in the MERRA-2 data set.

(2) As also mentioned by Reviewer 1, it would be desirable to have better constrained reanalysis data in the stratopause region. Starting with the year 2004 MERRA-2 assimilates MLS data. For this later period MERRA-2 should be one of the best freely available reanalysis data sets in the stratopause region. Therefore it would be interesting to additionally perform the MERRA-2 analysis only for the period starting with winter 2004/2005 to find out whether there is an effect on the MERRA-2 results, and whether the agreement between lidar and MERRA-2 climatologies changes and perhaps even improves.

The reviewer is right. It is of cause desirable to have better data in the stratopause region and that the assimilation of MLS data to MERRA-2 starting in 2004 is improving the data quality of that region (Galero et al., 2017; e.g., their Figure 21). As suggested we performed our MERRA-2 analysis also for the period starting in 2004/05. Firstly, we compare the vertical profiles of the climatology and mean STE from the whole data set with the period starting in winter 2004/05 (see Figure 1 below) for both locations. In Andenes the stratopause altitude of the climatological vertical profile is about 5km lower in the MLS period (blue dashed line) than in the whole MERRA-2 period (blue solid line) which fits much better to the lidar observations (green dashed line). In Kühlungsborn the stratopause altitude is only slightly decreased by 2km in the MLS period compared to the whole available period which again approximates the MERRA-2 data closer to the lidar observation. The largest differences appear above 45km for both locations which is the altitude region where MLS data are assimilated. Only the profiles for Andenes also show differences between 35 and 45km. Thus the assimilation of MLS data definitely improves the upper stratosphere and lower mesosphere in the MERRA-2 data. The profiles of the mean STE are almost equal between both periods and in both locations. Only the maximum values around the stratopause are about 3K higher in the MLS period than in the completely available period.

[Figure]

*Figure 1: Vertical profiles of the climatologies (blue) based on the whole MERRA-2 data set (solid) and on the period starting with winter 2004/05 (dashed) as well as the mean STE profiles (red) again based on the complete MERRA-2 data set (solid) and on the period starting with winter 2004/05 (dashed). The green dashed line represents the lidar climatology.*

The altitude of the STE definition is slightly different between lidar and MERRA-2. While for the lidar data the actual stratopause height is used, a fixed pressure level (2hPa) is used for MERRA-2 data. The reason for this difference is that there is only a low number of lidar measurements with strongly varying stratopause heights. At the same time, we want to compare our results with other studies that's why we take the fixed pressure level for MERRA-2 data. We clarified the differences in the "Method" subsection P4L22-25.

This difference in the STE definition explains large parts of the STE occurrence rates differences. In the table below you can find the different STE occurrence rates derived from MERRA-2 for different periods and STE definitions heights. Besides a period-to-period variability there is a general higher STE occurrence rate in the stratopause definition than in the fixed pressure level definition. The STE occurrence rate in the lidar period derived from MERRA-2 is identical to the lidar occurrence rate at Andenes and still slightly lower at Kühlungsborn. The remaining differences can be attributed to the different statistical basis between lidar and MERRA-2 and to not well represent small-scale dynamics like gravity waves in MERRA-2. We added a more detailed discussion of this topic in the paper on P8L11-24. Note that the two STE definition heights result in similar mean STE profiles at both location but with a slightly higher peak altitude (not shown).

| Kühlungsborn | 1980/81 – 2018/19 | | 2001/02 – 2011/12 | | 2004/05 – 2018/19 | |
|---|---|---|---|---|---|---|
| # all available days | 3510 | | 990 | | 1350 | |
| | 2hPa | Stratopause | 2hPa | Stratopause | 2hPa | Stratopause |
| # STE | 601 | 842 | 210 | 230 | 226 | 245 |
| STE occurrence rate | 17% | 24% | 21% | 23% | 17% | 18% |

| Andenes | 1980/81 – 2018/19 | | 2001/02 – 2011/12 | | 2004/05 – 2018/19 | |
|---|---|---|---|---|---|---|
| # all available days | 3510 | | 990 | | 1350 | |
| | 2hPa | Stratopause | 2hPa | Stratopause | 2hPa | Stratopause |
| # STE | 498 | 950 | 157 | 296 | 178 | 337 |
| STE occurrence rate | 14% | 27% | 16% | 30% | 13% | 25% |

**Specific comments:**

(1) p.2, l.2: The reference Hitchman et al. (1989) should be mentioned also here because this is the earlier work.

Done

(2) p.2, l.5/6: Did you mean the variability of the stratopause, here? Or the mean state? Or both?
Here we meant the variability of the stratopause region apart from the climatological mean state. We clarified that in the text on P2L5/6.

(3) p.4, l.5-10: Please add the information that, different from most other reanalyses, MERRA-2 assimilates MLS satellite soundings in the stratosphere and the mesosphere, starting with 2004. Therefore stratopause temperatures in MERRA-2 should be more reliable after that.
We thank the reviewer for this information. This emphasizes our choice to use MERRA-2. We added two sentences on this in the text P4L10-13.

(4) p.4, l.23: As averages as short as 3 hours are used for the lidar data also long period gravity waves may not completely average out. This may add some noise to the lidar STE statistics.
We thank the reviewer for this comment. We added long period gravity waves as a possible impact on the lidar data in the text (P4L29).

(5) Fig.2 seems to be in disorder: Firstly, the panels are not labeled (a) ... (d). Secondly, the caption is not correct: All nighttime measurements are displayed in gray, not blue, and their mean in blue, not red. Thirdly, it looks like the same average STE curve (in red) is used for both Andenes and Kühlungsborn. This does not make much sense because there are temperature offsets between both stations. Fourthly, the legend in (a) and (b) it should read "single soundings", not "single STE". Fifthly, please use English titles for the different panels.
Done

(6) p.5, l.13-16: Please note that in Fig.2 the same red curve is by mistake used in all panels. Therefore please check numbers after correction!
We thank the reviewer for finding this mistake. We have redone Figure 2 and checked all numbers. Numbers that changed are marked in red in the new manuscript. However the differences are only marginal mostly coming from the above mentioned bug in the data reading routine because a former version of the figure was correct where the numbers were taken from but shortly before submitting the manuscript we replaced this correct version by the wrong one. We are sorry for that.

(7) p.6 after l.22: You should add some discussion on the differences between MERRA-2 and lidar STE occurrence rates. The STE occurrence rate of MERRA-2 at the two locations is 14% and 17%, while for lidar these rates are 41/134=31% and 20/71=28%, about twice as high. This difference is quite remarkable and shows that large scale dynamics like planetary waves that are well represented in MERRA-2 explain about 50% of the STEs, which is an encouraging result, but are likely not the only mechanism causing STEs. Obviously, also smaller scale dynamics like gravity waves contribute to the lidar STE statistics.
We agree with the reviewer. Planetary waves dominate the STE development but gravity waves have also an impact. This impact seems to be higher in Kühlungsborn at mid latitudes than at Andenes at high latitudes (new Fig. 4). We discuss the difference between lidar and MERRA-2 STE occurrence rates now on P8L11 – P9L2 and P12L19-28 and for the impact of gravity waves on P12L30-P13L3.

(8) p.6, l.27: Here you write: "Figure 4 shows the original time series at one location at 2 hPa (blue line), ..." In the caption of Fig.4 it is mentioned that the blue line would be the "temperature deviation from the climatological daily mean". This information should also be given in the text of the manuscript. It should also be mentioned in the text that MERRA-2 data are shown.
We added both information regarding the description of Figure 4 (now Figure 5) to the text (P9L7 and P9L10).

(9) p.8, l.21/22: As has been shown in Fig.4, most of the temporal evolution of temperature anomalies can be described by a superposition of planetary waves, including the STE events. This sparks the question whether "localized momentum forcing resulting in a synoptic ageostrophic circulation" is needed to explain the STEs. It seems that STEs can be caused already by superposition of conservatively evolving planetary waves.
The more complex explanation of STE formation is discussed to show the reader that other studies found a different explanation. However, as proposed by the reviewer, we now relate this discussion to our results on P12L9 – 11.

(10) p.8,p.9: In the Discussion and the Summary it should be mentioned that the STE occurrence rates derived from lidar are twice as high as for MERRA-2 which indicates that dynamics at smaller scales can also lead to STEs.
For a more detailed answer please see above. We added the different STE occurrence rates of lidar and MERRA-2 in the text (P6L3/4 and P8L4/5) and in the Summary (P13L24/25) and discussed it in more detail on P8L11-24.

**Other comments:**
(1) p.1, l.1: During winter the circulation -> During winter the circulation at mid and high latitudes
Done
(2) p.2, l.17: in the northern hemisphere winter -> in the northern hemisphere during the winter season
Done
(3) p.2, l.33: The word "whereby" does not fit here. Suggest to start new sentence: (54N, 11E) whereby Andenes... -> (54N, 11E). Andenes...
Done
(4) p.3, l.1: winter -> winters
Done
(5) p.3, l.1: structure -> structured
Done
(6) p.4, l.9: winter -> winters
Done
(7) p.4, l.18: measuerement -> measurement
Done
(8) p.4, l.32: in phase of the wave. -> in phase of the wave is taken into account.
Done
(9) p.4, l.32/33: suggested rewording for the sake of comprehensibility: For example, the amplitude of a PW remains the same but changes in phase and thus can change the value of the local displacement at Andenes. -> For example, the amplitude of a PW could remain the same, but changes in phase still can change the value of the local displacement at a given location.
Done
(10) p.5, l.12: (red line) -> (blue line)
Done
(11) p.5, l.19: the climatology -> the temperature climatology
Done
(12) p.5, l.19: winter -> winters
Done
(13) caption of Fig.3: the long-term mean is subtracted. -> the long-term mean (blue line) is subtracted.
Done
(14) p.8, l.5: polar stratopause -> polar winter stratopause
Done

(15) caption of Fig.3: Külungsborn -> Kühlungsborn

Done